# Fungistatic Activity Mediated by Volatile Organic Compounds Is Isolate-Dependent in *Trichoderma* sp. “*atroviride* B”

**DOI:** 10.3390/jof9020238

**Published:** 2023-02-10

**Authors:** Eline van Zijll de Jong, Janaki Kandula, Michael Rostás, Diwakar Kandula, John Hampton, Artemio Mendoza-Mendoza

**Affiliations:** 1Bio-Protection Research Centre, Lincoln University, Lincoln 7647, New Zealand; 2Linnaeus Laboratory Ltd., Gisborne 4010, New Zealand; 3Agricultural Entomology, Department of Crop Sciences, University of Göttingen, 37077 Göttingen, Germany; 4Faculty of Agriculture and Life Sciences, Lincoln University, Lincoln 7647, New Zealand

**Keywords:** *Trichoderma* sp. “*atroviride* B”, volatile organic compounds, biocontrol, fungistatic activity

## Abstract

*Trichoderma* spp. produce multiple bioactive volatile organic compounds (VOCs). While the bioactivity of VOCs from different *Trichoderma* species is well documented, information on intraspecific variation is limited. The fungistatic activity of VOCs emitted by 59 *Trichoderma* sp. “*atroviride* B” isolates against the pathogen *Rhizoctonia solani* was investigated. Eight isolates representing the two extremes of bioactivity against *R. solani* were also assessed against *Alternaria radicina*, *Fusarium oxysporum* f. sp. *lycopersici* and *Sclerotinia sclerotiorum*. VOCs profiles of these eight isolates were analyzed using gas chromatography–mass spectrometry (GC-MS) to identify a correlation between specific VOCs and bioactivity, and 11 VOCs were evaluated for bioactivity against the pathogens. Bioactivity against *R*. *solani* varied among the fifty-nine isolates, with five being strongly antagonistic. All eight selected isolates inhibited the growth of all four pathogens, with bioactivity being lowest against *F*. *oxysporum* f. sp. *lycopersici*. In total, 32 VOCs were detected, with individual isolates producing between 19 and 28 VOCs. There was a significant direct correlation between VOC number/quantity and bioactivity against *R*. *solani*. 6-pentyl-α-pyrone was the most abundant VOC produced, but 15 other VOCs were also correlated with bioactivity. All 11 VOCs tested inhibited *R. solani* growth, some by >50%. Some of the VOCs also inhibited the growth of the other pathogens by >50%. This study demonstrates significant intraspecific differences in VOC profiles and fungistatic activity supporting the existence of biological diversity within *Trichoderma* isolates from the same species, a factor in many cases ignored during the development of biological control agents.

## 1. Introduction

Filamentous fungi of the *Trichoderma* genus are cosmopolitan organisms existing from marine to desert ecosystems [1,2,3,4,5]. *Trichoderma* spp. are either free-living or establish associations with other organisms, including bacteria, nematodes, other fungi, plants, and marine sponges [6,7,8,9,10,11]. These associations embrace diverse lifestyles ranging from mutualistic to parasitic [6,11]. This diversity in *Trichoderma* lifestyles might be related to their capacity to secrete a vast range of secondary metabolites and cellulolytic enzymes [11,12,13]. The capacity to produce these molecules, in addition to their ease of propagation, has made *Trichoderma* one of the most extensively used fungi in agriculture, where they are used as biofertilizers, biocontrol agents or plant protectants [14]. 

Currently, 375 *Trichoderma* species distributed worldwide have been described [15]. In New Zealand of 71 species identified, *Trichoderma* sp. “*atroviride* B”, a sister species to *T. atroviride* s.s., was the most commonly recovered species [16]. Contrary to the extensive distribution of *T. atroviride s.s.*, isolates of *T.* sp. “*atroviride* B” have to date only been discovered in the southern hemisphere, including New Zealand, Australia, South Africa and South America [16]. *T.* sp. “*atroviride* B” has been isolated from diverse environments and hosts, including wood, soil, the rhizosphere and other fungi. Several *T.* sp. “*atroviride* B” isolates protect plants by direct and indirect mechanisms [13,17]. These attributes have been used in agriculture to control soil-borne fungal pathogens (e.g., *Rhizoctonia solani* [18], and *Sclerotinia cepivorum* [19]), or foliar pathogens (e.g., *Botrytis cinerea* [20]). *T.* sp. *“atroviride* B*”* isolates also have plant growth promotion activity [21] and induce systemic resistance against foliar bacterial pathogens such as *Pseudomonas syringae* [22]. 

Microbial volatile organic compounds (VOCs) are chemically diverse small molecules with a low evaporation point. Due to their physical properties, VOCs are chemical signals that have an action far from the place of production [23,24,25,26]. *Trichoderma* spp. produce multiple VOCs with solid activity for plant growth promotion, fungistatic activity and abiotic stress protection [13,27,28,29,30,31]. The blend of VOCs released by twelve different *Trichoderma* species in *Arabidopsis* seedlings suggested a diversity in plant growth responses ranging from positive to no effect to negative [28]. *Trichoderma* isolates from the same species showed differences in plant growth induction, a mechanism linked with the blend of VOCs released by the fungi [13,32]. For example, VOCs released by *Trichoderma asperellum* IsmT5 had a negative impact on plant growth [32], while VOCs emitted by *T. asperellum* LU1370 resulted in a positive plant growth induction [13]. Similarly, some species of *Trichoderma* showed different VOCs profiles and antagonism toward the ectomycorrhiza *Laccaria bicolor* [33]. However, a systematic analysis of the fungistatic activity controlled by VOCs emitted by isolates from the same *Trichoderma* species is lacking. The information generated in this study will be relevant for understanding the mechanisms and molecules linked to the diversity in activity mediated by VOCs in *Trichoderma*. To develop a *Trichoderma*-based generic bio-inoculant, isolates from the same *Trichoderma* species from various locations within New Zealand were screened. A cohort of four isolates belonging to *T.* sp. *“atroviride* B*”* provided promising biological control and/or growth promotion when tested in a range of host/pathogen systems [34,35].

Here we report a systematic analysis of the fungistatic activity of VOCs emitted by fifty-nine *T.* sp*. “atroviride* B” isolates, and the ability of VOC blends of eight of these isolates showing distinctive fungistatic activity against *R. solani*. We also evaluated the effect of these individual VOCs on *R. solani*, *Sclerotinia sclerotiorum*, *Alternaria radicina* and *Fusarium oxysporum* f. sp*. lycopersici*. 

## 2. Materials and Methods 

### 2.1. Biological Materials

The 59 isolates of *T.* sp. “*atroviride* B” were from the Lincoln University Culture Collection and are of New Zealand origin [16]. Four of these isolates, referred to as the patented strains in Table 1, are patented for the biological control of soil-borne plant pathogens and for promoting plant growth. Sporulating agar discs of each isolate were cultured on prune extract agar [36] at 20 °C under blue light, then stored at 4 °C in sterile reverse osmosis water. *Rhizoctonia solani* RsS73 isolate (hereafter referred to as *R. solani*) from perennial ryegrass and *Sclerotinia sclerotiorum* from oilseed rape are New Zealand isolates and were obtained from the Lincoln University Culture Collection. *Alternaria radicina* (ICMP 10124) and *Fusarium oxysporum* f. sp. *lycopersici* (ICMP 5204) were obtained from the International Collection of Microorganisms from Plants (Landcare Research, Lincoln, New Zealand). These pathogens were maintained at 4 °C after sub-culture every 1–4 weeks at 25 °C in the dark on 2.4% (*w*/*v*) potato dextrose agar (PDA; Difco, BD, Franklin Lakes, NJ, USA).

### 2.2. Inverted Plate Assays

Aliquots (10 μL) of spore suspension from each *Trichoderma* isolate were grown in separate 90 × 15 mm vented polystyrene (PS) Petri dishes (Thermo Fisher Scientific Inc. Waltham, MA, USA) containing 15 mL of 2.4% (*w*/*v*) PDA at 25 °C in the dark for 2 days. The pathogens were grown similarly from a mycelial disc (6 mm diameter); *S. sclerotiorum* for 2 days, *R. solani* for 3 days, *F. oxysporum* f. sp. *lycopersici* for 9 days, and *A. radicina* for 11 days. Occasionally the cultures were stored at 4 °C for 1–3 days before inoculation of the assays.

The inverted plate assays were carried out in 90 × 25 mm vented PS Petri dishes containing 15 mL of 2.4% (*w*/*v*) PDA. A mycelial disc (6 mm diameter) was transferred from the periphery of the *Trichoderma* culture to the center of a Petri dish. After 0 or 48 h at 25 °C in the dark, a mycelial disc was transferred from the periphery of the pathogen culture to a new Petri dish and placed inverted over the *Trichoderma* culture. The Petri dishes were sealed together with a triple layer of plastic film and incubated at 25 °C in the dark. The controls were prepared in a similar manner except that a disc of PDA was used to inoculate the lower Petri dish in the positive control and both Petri dishes in the negative control.

### 2.3. Evaluation of Trichoderma Isolates

Two experiments were conducted to investigate the bioactivity of the fifty-nine isolates of *T.* sp. “*atroviride* B” in the inverted plate assays. *R. solani* was introduced at the same time as *Trichoderma* in the first experiment, and the assays were arranged as a randomized complete block (RCB) design with three blocks at 25 °C in the dark. The positive control was replicated eight times in each block, as were the four patented strains of *T*. sp. “*atroviride* B”, to minimize the variance of the difference between these and the other treatments. The second experiment was carried out similarly, except there were five blocks, and the pathogen was introduced 48 h after *Trichoderma*.

Twenty-one days after the pathogen was introduced, *Trichoderma* was removed and replaced with a Petri dish lid. After sealing with a triple layer of plastic film, the cultures of *R. solani* were incubated for a further 14 days to determine whether the effect of *Trichoderma* was fungicidal or fungistatic.

The diameter of *R. solani* colonies was recorded at their widest point every day from 0 to 6 or 7 days post-inoculation (DPI) and thereafter weekly up to 21 or 35 DPI, respectively. Due to the upward curve of the PDA at the edge of the Petri dish, it was not possible to accurately measure the diameter of the colony beyond 80 mm; hence this measurement was taken as the maximum colony diameter. The measurements were used to calculate the percentage inhibition of pathogen growth by the *Trichoderma* treatments compared to the positive control. The average inhibition of pathogen growth was determined by calculating the area under the curve following the trapezoid rule and dividing by the number of days between the first and last assessment. The first appearance of spores or sclerotia or spores were recorded for *Trichoderma* and *R. solani*, respectively, at the same time intervals that the diameters were measured.

ANOVA was performed on the percentage inhibition of pathogen growth on average and at each time point and the number of inhibition days. Chi-squared tests were conducted to test for significant differences in the occurrence of sclerotia between treatments and the positive control. These, together with the days of inhibition, were transformed before analysis using the base 10 logarithm to reduce the differences in variance between those with scores occurring when assays were assessed daily and those when assays were assessed weekly. Treatments with means equal or close to the minimum or maximum value were omitted from the analysis to avoid violating the assumption of equal variance, and the treatment structure was adjusted accordingly. These treatments were statistically compared to the variable treatments using the least significant effect (LSEffect 5%), that is the least significant difference (LSD 5%) divided by the square root of 2.

The bioactivity of eight isolates of *T.* sp. “*atroviride* B” (LU132, LU140, LU521, LU583, LU584, LU633, LU657, and LU661) was also evaluated in the inverted plate assays against three other pathogens; *A. radicina*, *F. oxysporum* f. sp. *lycopersici* and *S. sclerotiorum* and compared to *R. solani* in duplicate experiments following a randomized complete block design with four blocks. The positive controls for each pathogen were replicated three times in each block. The assays were conducted at 25 °C in the dark, and the pathogens were introduced 48 h after *Trichoderma*.

The maximum colony diameter and colony diameter perpendicular to the maximum were recorded for each pathogen when they were, on average, ≥65 mm in the positive control (1–2 DPI for *S. sclerotiorum*, 2 DPI for *R. solani*, 7 DPI for *F. oxysporum* f. sp. *lycopersici* and 14 DPI for *A. radicina*). The percentage inhibition of pathogen growth was calculated based on the average of these two measurements. ANOVA was performed for a randomized complete block design with four blocks and a factorial treatment structure of 4 (pathogen) × 8 (*Trichoderma*). A combined analysis was also performed using the data means from the two experiments.

### 2.4. Analysis of VOCs

The VOC profiles of eight isolates of *T.* sp. “*atroviride* B” (LU132, LU140, LU521, LU583, LU584, LU633, LU657, LU661) were analyzed in the inverted plate assays by gas-chromatography–mass spectrometry (GC-MS). The assays were carried out at 25 °C in the dark. *R. solani* was introduced 48 h after *Trichoderma*, and a blue PTFE silicone septum (MicroAnalytix NZ Ltd, Auckland, New Zealand) was placed over a hole (3 mm diameter) present along the top edge of the Petri dish with *Trichoderma* before the Petri dishes were sealed together with five layers of plastic film (Figure 1). The experiment was carried out as an RCB design with five blocks. Each block was prepared on a different day. The set-up of each treatment within a block was staggered to take into account the run-time on the GC-MS so that the VOCs were sampled from each treatment at the same stage of growth.

The VOCs were extracted 48 h after *R. solani* inoculation from the headspace by solid-phase micro-extraction (SPME) using 10 mm of a 65 μm polydimethylsiloxane/divinylbenzene (PDMS/DVB) fiber with 23 gauge needle (Supelco, Sigma-Aldrich New Zealand Co., Auckland, New Zealand ) at 25 °C for 25 min. The Petri dishes were clamped perpendicularly for sampling. The diameter of the pathogen colony was recorded at its widest point immediately after sampling and used to calculate the percentage inhibition of pathogen growth (Figure 1).

The VOCs were analyzed on a GCMS-QP2010 Ultra (Shimadzu Scientific Instruments, Auckland, New Zealand) using an Rtx-5MS column (diphenyl dimethylpolysiloxane, 30 m × 0.25 mm i.d., 0.25 μm film thickness, Restek). They were desorbed from the fiber by splitless injection at 250 °C for 2 min, and after an additional 8 min, the fiber was removed. Helium was used as the carrier gas at a flow rate of 1.0 mL/min. The column oven temperature was held at 40 °C for 2 min before being raised to 200 °C at a rate of 10 °C/min and then to 260 °C at a rate of 25 °C/min, where it was held for 5 min. The interface temperature was 260 °C and the ion source was 230 °C. The detector voltage was 0.85 kV, and the scanning range was 33–500 m/z. An aliquot (5 μL) of alkane mix (C8–C20, Sigma-Aldrich) was placed in a 20 mL amber glass headspace vial and sealed with a screw cap containing a blue PTFE/silicone septum (Sigma-Aldrich) which was sampled and analyzed as described above. Standards (2 mg/L) were sampled by liquid injection and analyzed on the GC-MS as described above.

Compounds with a peak with a minimum slope of 300/min were identified by comparison of their mass spectra with those in the NIST11 and Wiley10 databases using the software GCMSsolution (version 4.11, Shimadzu). Those compounds with identity to siloxane were removed. Likewise, compounds present in the negative (uninoculated PDA plates) and positive (PDA plates inoculated with *R. solani*) controls were also removed unless the peak areas in ≥5 treatments were significantly higher (*p* < 0.05) than the controls. The percentage similarity to the compound was recorded for the peak with the highest peak area in each block. The retention index (RI) was determined for these peaks using the software MassFinder (version 4.26, Dr Hochmuth Scientific Consulting). The identity of those with a retention index that did not match the published retention index for the compound (accessed on 15 May 2018, NIST Chemistry WebBook: http://webbook.nist.gov/chemistry/) was amended to unknown. The base peak chromatogram was also examined, and those with more than one base peak were de-convoluted using the software AMDIS (version 2.71, http://chemdata.nist.gov/mass-spc/amdis/downloads/, accessed on 15 May 2018). Compounds were then matched within and between the five blocks in the RCB design based on their identity and retention index or retention time. Only those that occurred in at least one treatment across ≥3 blocks were included in the statistical analysis.

To estimate the linear range of VOCs, a calibration curve from five compounds that occurred in the fungal headspaces was calculated according to Bartelt (1997). The VOCs 2-heptanone, 6-pentyl-α-pyrone, (+)-limonene, undecane and tridecane (for purity, see below) were dissolved in *n*-hexane. Each mixture was added to a 20 mL SPME glass vial in four concentrations (1, 2.5, 10, and 25 µg L^−1^) and analyzed in four technical replicates by GC-MS as described above. 

ANOVA was performed on the percentage inhibition of pathogen growth and peak area of a compound. The correlation between the total number of VOCs or peak area of VOCs and the percentage inhibition of pathogen growth was tested by linear regression using the treatment means. Likewise, a linear regression was performed for each VOC using the treatment means for the peak area and percentage inhibition of pathogen growth to identify those displaying a significant positive correlation with bioactivity.

### 2.5. Bioactivity Assays of Pure Compounds

A total of eleven VOCs identified from GC-MS analysis were evaluated for bioactivity against *R. solani* and three other pathogens (*A. radicina*, *F. oxysporum* f. sp. *lycopersici* and *S. sclerotiorum*) in inverted plate assays. The compounds farnesene (mixture of isomers), geranylacetone (≥97%, CAS no. 689-67-8), (+)-limonene (97%, CAS no. 5989-27-5), (-)-limonene (96%, CAS no. 5989-54-8), nerolidol (mixture of cis and trans, ≥97%, CAS no. 7212-44-4), 6-pentyl-α-pyrone (≥96%, CAS no. 27593-23-3), tridecane (≥99%, CAS no. 629-50-5), undecane (≥99%, CAS no. 1120-21-4) and 2-undecanone (≥98%, CAS no. 112-12-9) were obtained from Sigma-Aldrich, bisabolene (CAS no. 17627-44-0) from Indukern, and β-curcumene (CAS no. 451-56-9) from Extrasynthese. 

The inverted plate assays were carried out as described previously [27]. The compounds (0.32, 1.6, 8, 40, 200 or 1000 μmol) were applied together with hexane (hexane (95%, Thermo Fisher Scientific) in 50 μL volumes to 1–5 antibiotic assay discs (13 mm diameter, Whatman, GE Healthcare Life Sciences, Auckland, New Zealand) adhered around the center of a Petri dish lid with dichloromethane (5 μL/disc, Applied Biosystems, Waltham, MA, USA). The Petri dish with the pathogen was placed inverted over the discs and sealed with plastic film. The positive control was exposed to 50 μL of hexane. The assays for each VOC were placed in separate incubators at 25 °C in the dark.

Each VOC was evaluated in duplicate experiments as a randomized complete block design with four blocks. The only exception was β-curcumene which, due to the cost of this compound, was only evaluated once. The highest quantity of each VOC (1000 μmol) was tested against all four pathogens, whereas the other quantities (0.32–200 μmol) were only tested against *R. solani*. For the latter pathogen, there were two positive controls per block.

The mean colony diameter was recorded for the four pathogens as described in Section 2.3 ANOVA was performed for each experiment using the base 10 logarithms of the mean colony diameters of the positive control over the treatment. A combined analysis was also performed using the data means from the two experiments.

Additional experiments were conducted with 6-pentyl-α-pyrone and the two stereoisomers of limonene to determine the quantity required for 50% growth inhibition of *R. solani*. These VOCs were tested in duplicate experiments against *R. solani* at quantities of 200, 360, 520, 680, 840 and 1000 μmol as described above.

## 3. Results

### 3.1. Bioactivity of VOCs from Trichoderma sp. “atroviride B” against R. solani 

The suppressive activity of VOCs emitted by *T*. sp. “*atroviride* B”, the four patented strains and additional isolates of *T*. sp. “*atroviride* B”, previously identified [16],were tested against *R. solani*. There were differences in the level and duration of *R. solani* inhibition among the four patented strains and other isolates of *T*. sp. “*atroviride* B” when the pathogen was introduced 48 h after *Trichoderma* in the inverted plate assays. Strain LU132 displayed significantly higher levels of inhibition than strains LU140, LU584 and LU633 and 32 other isolates (Table 1). Five isolates (LU521, LU532, LU661, LU723 and LU1308) were more antagonistic towards *R. solani* than strain LU132 (Figure 2a). Inhibition with strain LU584 was significantly lower and shorter in duration than LU132, LU140 and LU633. There were no isolates that, on average, displayed significantly lower levels of inhibition than LU584, but inhibition was lower at two or more time points with four isolates; LU524, LU583, LU657 and LU658 (Figure 2b). The duration of antagonism was shorter for isolate LU657. None of the 59 *Trichoderma* isolates inhibited *R. solani* growth when both fungi were introduced simultaneously (data not shown).

Fifty-two of the *Trichoderma* isolates significantly reduced the occurrence of sclerotia in pathogen cultures (Table 1). For three isolates that strongly inhibited pathogen growth (LU661, LU723 and LU1308), the observed reduction was not significantly lower than the positive control. Some variation was detected in the growth of *Trichoderma* isolates. Unlike the other isolates, cultures of isolate LU499 were not white and fluffy in appearance. The growth rates of some of the best (LU532 and LU723) and worst (LU657) performing isolates were slower than that of the patented strains (Table 2). Two of the isolates with low bioactivity (LU524 and LU658) sporulated earlier than the four patented strains, whereas another (LU657) sporulated later, as did one of the isolates with high bioactivity (LU661). One isolate (LU723) had not sporulated by 22 days.

### 3.2. Bioactivity of VOCs from T. sp. “atroviride B” against other Pathogens 

The four patented strains and the best and worst-performing isolates (as evaluated against *R. solani*) were able to inhibit the growth of other pathogens in the inverted plate assays (Figure 3). The best-performing isolate, LU521, showed high levels of bioactivity against *R. solani*, *A. radicina* and *S. sclerotiorum*, whereas bioactivity was lower against *F. oxysporum* f. sp. *lycopersici*. This isolate displayed significantly higher levels of bioactivity against these pathogens than most of the other *Trichoderma* isolates.

The four patented strains (LU 132, LU140, LU584 and LU633) were more antagonistic towards *R. solani* than the other pathogens (Figure 3). Inhibition of *A. radicina* and *S. sclerotiorum* was significantly higher than of *F. oxysporum* f. sp. *lycopersici*. The different strains displayed similar levels of bioactivity, and only strain LU633 tended to be less antagonistic than the other strains towards *A. radicina*. 

### 3.3. Analysis of VOCs from T. sp. “atroviride B”

A total of 32 VOCs detected in the headspace of the inverted plate assays were categorized as having *Trichoderma* origin (Table 3). These were not detected on PDA plates without fungi (negative control) or on plates containing *R. solani* alone. *Trichoderma* isolates produced, on average, between 19 and 28 VOCs (Table 3). There was a direct correlation between the total number and quantity of VOCs produced by an isolate and bioactivity against *R. solani* (*r* = 0.93 and *r* = 0.87, respectively, *p* < 0.001).

The most abundant VOC produced by all *Trichoderma* isolates, 6-pentyl-α-pyrone (Table 4) (VOC 21), displayed a significant positive correlation with inhibition of *R. solani* growth (Figure 4). The abundance of 15 other VOCs was also correlated with bioactivity. These displayed identities to the compounds undecane (VOC 8), tridecane (VOC 15), (±)-limonene (VOC 5), trans-nerolidol (VOC 28), β-bisabolene (VOC 25), cis-β-farnesene (VOC 20), (−)-germacrene D (VOC 22), α-bergamotene (VOC 18), 2-undecanone (VOC 14) and geranylacetone (VOC 19). The identities of the remaining VOCs (VOC 9, 23, 24, 30 and 32) were unknown.

Isolate LU521, which displayed significantly higher levels of bioactivity (Table 4), produced three VOCs that were low or absent in the other *Trichoderma* isolates (Table 5). The identity of one was unknown (VOC 32), but the remaining two showed identity to 2-undecanone (VOC 14) and Methyl trans,cis-farnesate (VOC 29). Three other VOCs, verticiol (VOC 31) and two unknowns (VOC 16 and 30), were also detected in high levels in this isolate and in isolate LU661 (Table 5).

Calibration curves of five fungal VOCs (2-heptanone, 6-pentyl-α-pyrone, (+)-limonene, undecane and tridecane) were created to estimate absolute VOC concentrations in the fungal headspace samples (Appendix A). Linear regression analyses resulted in R^2^ values between 0.9298 for 2-heptanone and 0.9965 for 6-pentyl-α-pyrone. However, the compounds differed considerably in their affinity to the SPME fiber, and the full range of concentrations was not covered. Hence, only relative peak areas are given for fungal headspace VOCs (Table 5).

### 3.4. Bioactivity of Individual VOCs

The growth of *R. solani* was inhibited by all 11 of the tested VOCs (Figure 4). Seven of the VOCs inhibited growth by >50%. This level of inhibition was reached with lower concentrations of nerolidol, 2-undecanone, and geranylacetone (Figure 4), than of 6-pentyl-α-pyrone, (+)-limonene and (−)-limonene (Figure 4). Between 63–89% inhibition was detected with these VOCs. Inhibition was highest with 2-undecanone. Lower levels of inhibition (25–38%) were detected with the other five VOCs; undecane, tridecane, β-curcumene, bisabolene and farnesene.

In addition to their activity against *R. solani*, the VOCs 6-pentyl-α-pyrone, nerolidol, 2-undecanone, and geranylacetone inhibited the growth of *A. radicina* by >50%, as did 6-pentyl-α-pyrone and 2-undecanone against *F. oxysporum* f. sp. *lycopersici*, and 2-undecanone, geranylacetone, (+)-limonene and (−)-limonene against *S. sclerotiorum* (Figure 5). Inhibition levels were similar to or lower than those detected in *R. solani*, except for 6-pentyl-α-pyrone, which was more antagonistic towards *A. radicina* (*p* < 0.01). (Figure 5).

## 4. Discussion

This study shows that the blend of fungal volatile organic compounds is highly variable within the same *Trichoderma* species and depends on the specific isolate. Differences were found in the number of VOCs emitted by *T*. sp. “*atroviride* B” isolates, ranging from 19 to 28. The VOCs also differed in their concentrations. The fungistatic and fungicidal activities of *Trichoderma* VOCs, using inverted plate bioassays, have previously been reported [27,37] and were confirmed in the present study. Remarkably, the observed variation in VOC emission is directly associated with the extent of fungistatic activity against the plant pathogen *R. solani*.

*T*. sp. “*atroviride* B” can affect plants positively in a direct manner, for instance, by the production of beneficial metabolites or proteins and indirectly by inhibiting plant pathogens [17]. Intraspecific differences in inhibiting phytopathogens and promoting plant growth are well known but these also depend on the plant and pathogen species tested in addition to the fungal isolate [13,21,38]. Furthermore, *Trichoderma* isolates differ in their capacities to colonize the rhizosphere of diverse plant species [39]. In this work, we tested the VOC blends emitted by 59 different isolates of *T*. sp. “*atroviride* B” on the fungistatic activity against *R. solani.* Our data support a continuum in the antagonistic effects of VOC blends, ranging from strong inhibition to no effects on *R. solani* growth (Table 1). To our knowledge, this is the first intraspecific comparison of fungistatic activity caused by a large range of *Trichoderma* isolates. Our observations support previous results where a blend of VOCs from different *Trichoderma* isolates from the same species differentially affected plant growth in *Arabidopsis thaliana* [28,32]. Some VOC blends inhibited plant growth while other isolates had a positive effect [13,28]. Earlier studies showed that various *T*. sp. “*atroviride* B” isolates also respond differently to environmental cues; for example, while *T*. sp. “*atroviride* B” LU132 produces high levels of indole-3-acetic acid in the presence of L-tryptophan in the medium, this was not the case in *T*. sp. “*atroviride* B” LU660 [13]. Whether the observed specific variation in VOC composition is further influenced by the presence of a pathogen, such as *R. solani* or its strains, needs to be assessed in future studies.

We found that the VOC blends of *T*. sp. “*atroviride* B” isolates differentially suppressed sclerotia formation in *R. solani*. Hence, this is the first report showing intraspecific variability in the inhibition of sclerotia by a *Trichoderma* species. Likewise, it was reported that VOCs emitted by endophytic bacteria inhibit sclerotia formation in *Sclerotinia sclerotiorum* [40], suggesting a common strategy in organisms to manipulate growth in antagonists by volatiles. The sclerotia are important for the survival of pathogens under unfavorable environmental conditions, and by inhibiting their formation, *Trichoderma* contributes to eliminating the pathogen population. In *R*. *solani*, sclerotia formation is regulated by reactive oxygen species (ROS) and trehalose [41]. Further work is required to identify if VOCs emitted by *Trichoderma* modulate ROS and trehalose metabolism in plant pathogenic fungi.

The mechanisms behind VOC production in *Trichoderma* are not entirely understood; however, chemical and physical signals, including nutrients, light, temperature, pH, development stage and associations with other organisms have been described as inducers of VOC production [42,43,44,45,46,47]. The role of some signaling components in the production of VOCs in *T. atroviride* has been reported. For example, in the absence of the NADPH Oxidase Nox2 [27], the Histidine Kinase Two-Component Response Regulator Skn7 [37] or the MAPK Tmk3 [45] from*T. atroviride* IMI206040 were unable to produce 6-pentyl-α-pyrone, the most abundant VOC in this isolate. However, this is not the case in the absence of the NADPH Oxidase Nox1, the NADPH Oxidase Regulator NoxR or the Histidine Kinase Two-Component Response Regulator Ssk1 [27,37]. The lack of the Histidine Kinase Two-Component Response Regulator Rim15, on the other hand, modifies the accumulation of VOCs in *T. atroviride* IMI206040 [37]. The role of these signaling components was studied in the interaction with *R. solani* and *Sclerotinia sclerotiorum* [27,37]. These studies illustrated that airborne signals communication is differentially modulated by these signaling components in *Trichoderma* and the outcome of the interaction is also plant pathogen dependent [27,37].

Further investigation is required to identify the mechanisms behind the volatile-mediated interactions between *R. solani* and *T*. sp. *atroviride* B. Variability in VOC production was previously observed in the two *T. atroviride* strains P1 and IMI 206040, which regulate the production of VOCs differently under the same light stimulus [43]. Interestingly, the same signaling components (the MAP kinase Tmk3) in *T. atroviride* P1 and IMI 206040 have different capacities to perceive or respond to the same stimulus [43]. Our observations were based on using the same stimuli in 59 different isolates from the same species, with a broad range of effects in *R. solani*. Moreover, the blend of VOCs produced by *Trichoderma* varies depending on the environmental conditions. For example, *T. atroviride* IMI206040 produced 28 VOCs when the PDA medium was used [37,43,44,45,46], but only 13 VOCs were detected when the MS medium was used [13,46]. Consequently, different effects on plant physiology were observed [46].

During biotic interactions, signaling mediated by VOCs is a bi-directional process. For example, *Fusarium oxysporum* releases VOCs, which *Trichoderma* recognizes*;* consequently, there is a response in the mycoparasite and the other way around [47]. The fact that we could not detect VOCs emitted by *R. solani* Rs73 grown alone (data not shown) is intriguing, as it has been reported that another isolate of *R. solani* emits 8 VOCs [48,49]. One reason for the lack of VOC emission found here could be the variability among *Rhizoctonia* isolates; another may be the use of different VOC sampling methods. In this study solid phase microextraction (SPME) sampling was employed while the previous reports used thermal desorption with Tenax as an adsorbent. Nevertheless, we do not rule out entirely that the *Trichoderma* VOCs may have influenced the production of VOCs in *R. solani* and those detected represent a blend of VOCs from both, *Trichoderma* and *Rhizoctonia.* However, most of the VOCs reported in this work correspond to those reported in *T*. sp. ‘ “*atroviride* B” LU132 grown on plant synthetic medium (MS) [13] and other species of *Trichoderma*, including *Trichoderma atroviride sensu stricto* grown alone [28,49,50,51,52].

More than 470 different VOCs have been reported from various *Trichoderma* spp. [13,28,43,44,45,46,47]. Here, we identified 28 VOCs produced by *T*. sp. “*atroviride* B” in the presence of *R. solani.* Most of these molecules have been previously identified, and some correspond to the “classical” VOCs reported in *T. atroviride sensu stricto* [48], including 6-pentyl-α-pyrone, 2-undecanone, and 2-hepatanone. These molecules have also been reported in other species such as *T. gamsii, T. harzianum, T. asperellum* and *T. viride*, but not in *T. reesei* and *T. virens* (Gv29.8) [13,28,52,53,54,55]. A clear correlation between the number of *T.* sp. “*atroviride* B” VOCs and the fungistatic activity against the pathogen suggests that the blend of VOCs emitted by the isolates might have a synergistic effect in inhibiting plant pathogens (Figure 3 and Table 2). This potential synergism became evident when individual VOCs were tested; some of them did not show the same fungistatic activity in the four pathogens tested, supporting the idea that the blend of these compounds might be essential for this activity. Interestingly, the production of ketones (e.g., 6-pentyl-α-pyrone, 2-undecanone, geranylacetone) showed strong inhibitory effects on *R. solani* growth (Figure 4 and Figure 5), which may be due to the general reactivity of the keto functional group. This contrasts with the relatively low inhibitory action of the compounds without a functional group (n-alkanes: undecane, tridecane; sesquiterpenes: curcumene, bisabolene, farnesene). The case of limonene (a monoterpene without a functional group) is peculiar: the compound showed inhibitory effects, though only at the highest concentration.

In addition to differences in the fungistatic activity among *T*. sp. “*atroviride* B” isolates, a pathogen’s susceptibility to particular *Trichoderma* VOCs is species-specific too. The fungistatic activity of the individual VOCs differed among the tested plant pathogens. For instance, fungistasis in *R. solani* decreased in the following order: 2-undecanone > 6-pentyl-α-pyrone > (−) limonene > geranylacetone > (+) limonene; for *A. radicina* it was: 6-pentyl-α-pyrone > 2-undecanone > nerolidol mixture > geranylacetone and for *F. oxysporum* we determined: 6-pentyl-α-pyrone > 2-undecanone = nerolidol mixture > geranylacetone. For *S. sclerotiorum* the order was: (−) Limonene = (+) Limonene > 2-Undecanone. These observations suggest that the effects of VOCs on the pathogens differ depending on the VOC produced, and the outcome may depend on the blend of the different VOCs. We suggest that screening isolates for VOC blends that have shown strong fungistatic activity may deliver promising candidates for *Trichoderma*-based bioinoculant development.

## 5. Conclusions

*Trichoderma* sp. “*atroviride* B” isolates produce qualitatively and quantitatively different VOC blends.There is a direct correlation between the amount of VOCs emitted by *T.* sp*. “atroviride B”* and the fungistatic activity against *R. solani*.The blend of VOCs produced by *Trichoderma* has stronger fungistatic activity than the single VOCs tested.Plant pathogens respond species-specifically to single *Trichoderma* VOCs and the whole blend.

## 6. Patents

The strains LU132, LU584, LU633 and LU140 are patented (US 8,394,623).

## Figures and Tables

**Figure 1 jof-09-00238-f001:**
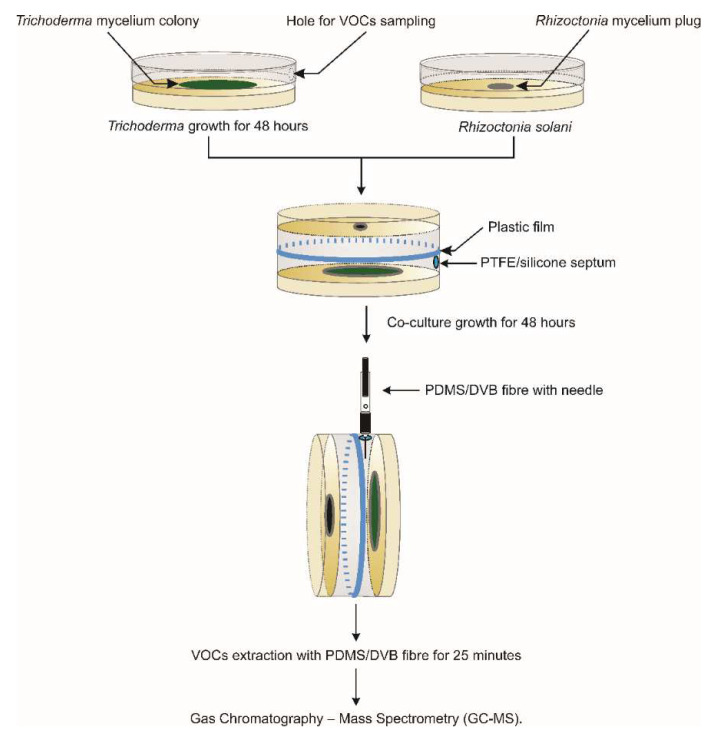
System for sampling volatile organic compounds produced by *Trichoderma* sp. “*atroviride* B” against *Rhizoctonia solani* in the inverted plate assay.

**Figure 2 jof-09-00238-f002:**
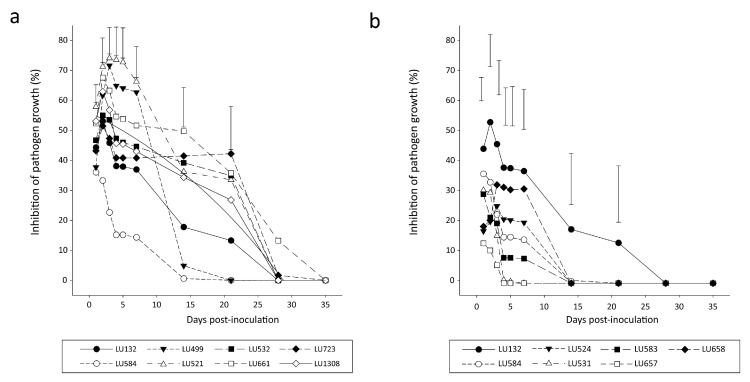
Inhibition of *Rhizoctonia solani* growth by the blend of volatile organic compounds emitted by two sets (**a**) and (**b**) of selected isolates of *Trichoderma* sp. “*atroviride* B” at various time points in the inverted plate assays. The pathogen was introduced 48 h after *Trichoderma*, and both were incubated at 25 °C, with the latter removed 21 days post-inoculation. The error bars indicate the least significant difference (LSD 5%) compared to isolates with the patented strains at the different time points with means >0. The experiment was conducted two times using five plates for each isolate.

**Figure 3 jof-09-00238-f003:**
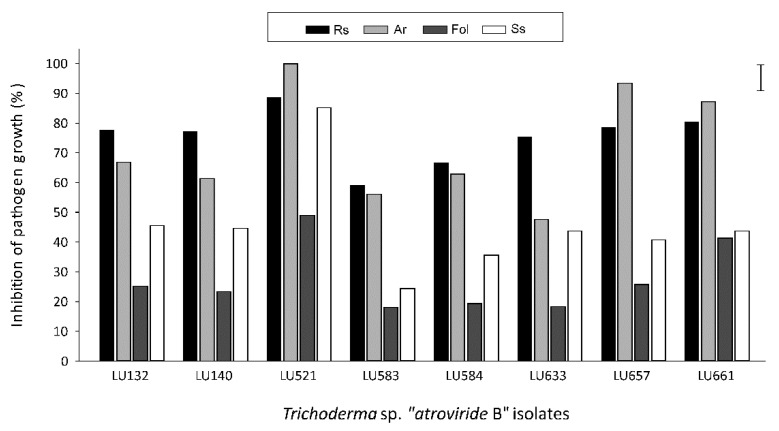
Mean inhibition of *Rhizoctonia solani* (Rs), *Alternaria radicina* (Ar), *Fusarium oxysporum* f. sp. *lycopersici* (Fol) and *Sclerotinia sclerotiorum* (Ss) growth by isolates of *Trichoderma* sp. “*atroviride* B” in the inverted plate assays. The pathogen was introduced 48 h after *Trichoderma,* and both were incubated together at 25 °C for 2 d with Rs, 14 d with Ar, 7 d with Fol and 1.4 d with Ss. The error bar indicates the least significant difference (LSD 5%). The means are from two independent experiments conducted following a randomized complete block design with four blocks.

**Figure 4 jof-09-00238-f004:**
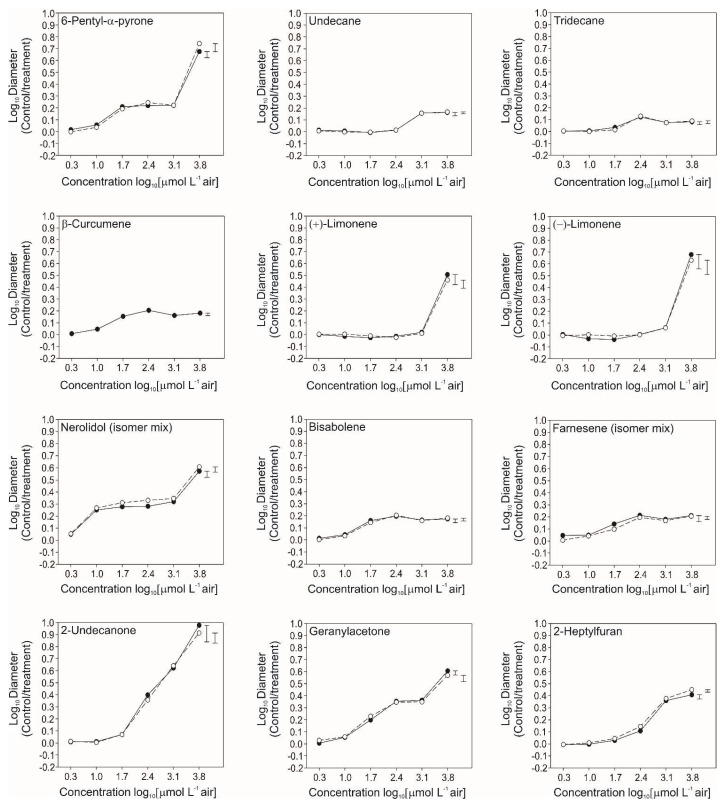
Mean inhibition of *Rhizoctonia solani* growth with individual volatile organic compounds (VOCs) identified in the headspace of the inverted plate assays with *Trichoderma* sp. “*atroviride* B” which displayed a positive correlation with bioactivity. The pathogen was incubated for 2 d at 25 °C above paper discs, to which one of six different concentrations of the VOC was applied at the time of pathogen inoculation. Closed circles correspond to Expt. 1, open circles correspond to Expt. 2. The error bars indicate the least significant difference (LSD 5%) for two independent experiments, Expt. 1 (left bar) and Expt. 2 (right bar). The values on the y-axis are from lowest to highest when back-transformed equivalent to −58, −26, 0, 21, 37, 50, 60, 68, 75, 80, 84, 87 and 90% inhibition of pathogen growth.

**Figure 5 jof-09-00238-f005:**
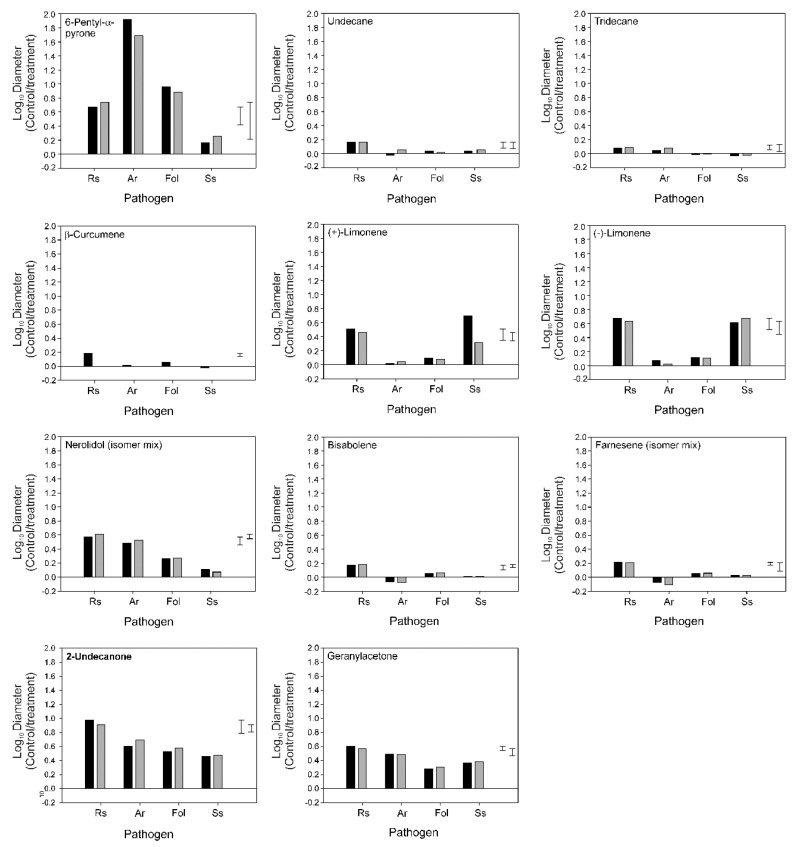
Mean inhibition of the growth of *Rhizoctonia solani* (Rs), *Alternaria radicina* (Ar), *Fusarium oxysporum* f. sp. *lycopersici* (Fol) and *Sclerotinia sclerotiorum* (Ss) by individual volatile organic compounds (VOCs) identified in the headspace of the inverted plate assays with *Trichoderma* sp. “*atroviride* B” that displayed a positive correlation with bioactivity. The pathogen was incubated at 25 °C for 2 d with Rs, 14 d with Ar, 7 d with Fol and 1.4 d with Ss above paper discs, to which the VOC was applied at a concentration of 3.84 log_10_ μmol L^−1^ air at the time of pathogen inoculation. The error bars indicate the least significant difference (LSD 5%) for two independent experiments, Expt. 1 (left bar) and Expt. 2 (right bar). Some of the VOCs were tested against Ss in a third experiment due to the abnormal growth of this pathogen in one of the earlier experiments. The values on the y-axis are from lowest to highest when back-transformed, equivalent to −58, 0, 37, 60, 75, 84, 90, 94, 96 and 97% inhibition of pathogen growth.

**Table 1 jof-09-00238-t001:** Average percentage inhibition of *Rhizoctonia solani* colony growth and occurrence of sclerotia in the presence of *Trichoderma* sp. “*atroviride* B” isolates in the inverted plate assays ^a^.

Strain	Replicates	Average Inhibition of Pathogen Growth (%) ^b^	Occurrence of Pathogen Sclerotia (%)	
Patented strains				
LU132	40	18.30	13.5	***^c^
LU140	40	14.40	36.8	***
LU584	40	5.75	25.0	***
LU633	40	14.43	18.9	***
Other isolates				
LU661	5	35.74	60.0	
LU521	5	33.61	0.0	***
LU723	5	30.44	50.0	
LU532	5	28.44	20.0	***
LU1308	5	25.91	60.0	
LU141	5	24.46	0.0	***
LU579	5	21.51	33.3	*
LU666	5	20.25	25.0	**
LU131	5	19.81	20.0	***
LU1345	5	19.10	50.0	
LU1357	5	18.82	20.0	***
LU499	5	18.51	40.0	*
LU522	5	18.28	40.0	*
LU665	5	17.51	0.0	***
LU572	5	17.23	20.0	***
LU659	5	17.13	60.0	
LU510	5	17.11	0.0	***
LU298	5	16.74	33.3	*
LU1341	5	15.78	25.0	**
LU1330	5	15.74	40.0	*
LU562	5	15.00	0.0	***
LU497	5	14.40	20.0	***
LU739	5	13.70	75.0	
LU660	5	13.32	25.0	**
LU634	5	12.29	20.0	***
LU668	5	11.89	40.0	*
LU575	5	11.02	50.0	
LU590	5	10.91	0.0	***
LU577	5	10.23	40.0	*
LU992	5	9.97	20.0	***
LU568	5	9.41	0.0	***
LU573	5	8.48	60.0	
LU658	5	8.24	0.0	***
LU136	5	8.21	20.0	***
LU563	5	7.99	50.0	
LU987	5	7.28	25.0	**
LU656	5	7.24	20.0	***
LU574	5	6.90	20.0	***
LU725	5	6.66	80.0	
LU578	5	6.45	40.0	*
LU300	5	6.34	20.0	***
LU587	5	6.21	20.0	***
LU147	5	6.02	20.0	***
LU588	5	5.98	40.0	*
LU591	5	5.93	60.0	
LU524	5	5.78	40.0	*
LU582	5	5.49	40.0	*
LU580	5	5.34	40.0	*
LU741	5	5.15	20.0	***
LU581	5	5.03	20.0	***
LU589	5	5.03	40.0	*
LU586	5	5.01	25.0	**
LU583	5	3.34	40.0	*
LU531	5	1.85	20.0	***
LU657	5	0.69	40.0	*
*R. solani* alone ^c^	40	N/A	92.5	
LSD (5%)	5 v. 5	8.41		
	5 v. 40	6.31		
	40 v. 40	2.97		

^a^ The pathogen was introduced 48 h after *Trichoderma,* and plates were incubated at 25 °C with the latter removed 21 days post-inoculation. The pathogen was grown for 35 days post-inoculation to detect sclerotia production. ^b^ Calculated from the average area under the curve of the percentage inhibition of pathogen growth by the *Trichoderma* strain from 0 to 35 days post-inoculation. ^c^ Significance of difference from *R. solani* growth alone in Chi-squared test where *: *p* ≤ 0.05; **: *p* ≤ 0.01; ***: *p* ≤ 0.001.

**Table 2 jof-09-00238-t002:** The growth and sporulation patterns of selected *Trichoderma* sp. “*atroviride* B” isolates in the inverted plate assays. *Rhizoctonia solani* was introduced 48 h after *Trichoderma* and these were incubated together at 25 °C for a further 21 days.

Treatment	Replicates	*Trichoderma* Growth (mm, Diameter) at 2 Days	Log10 Days for *Trichoderma* Colonization (Back Transformed)	Log10 Days for *Trichoderma* Sporulation (Back Transformed)
Patented strains				
LU132	40	(74.0) ^a^	(0.30) [2.0]	0.64 [4.4]
LU140	40	(74.0)	(0.30) [2.0]	0.63 [4.3]
LU584	40	(74.0)	(0.30) [2.0]	0.62 [4.2]
LU633	40	(74.0)	(0.30) [2.0]	0.60 [4.0]
Top performers				
LU499	5	(74.0)	(0.30) [2.0]	1.36 [22.9]
LU521	5	(74.0)	(0.30) [2.0]	0.64 [4.4]
LU532	5	40.2	0.67 [4.6]	0.64 [4.4]
LU661	5	(74.0)	(0.30) [2.0]	0.75 [5.6]
LU723	5	41.2	0.58 [3.8]	NS ^b^
LU1308	5	(74.0)	(0.30) [2.0]	0.66 [4.6]
Poor performers				
LU524	5	(74.0)	(0.30) [2.0]	0.48 [3.0]
LU531	5	18.0	1.10 [12.7]	0.66 [4.6]
LU583	5	(74.0)	(0.30) [2.0]	0.60 [4.0]
LU657	5	15.4	0.82 [6.6]	0.97 [9.3]
LU658	5	(74.0)	(0.30) [2.0]	0.53 [3.4]
LSD (5%)	5 v. 5	5.1	0.16	0.05
	5 v. 40	3.8	0.12	0.04
	40 v. 40	1.8	0.06	0.02
LSEffect (5%)	5 v. 5	3.6	0.11	
	5 v. 40	2.7	0.08	
	40 v. 40	1.3	0.04	

^a^ Numbers in round brackets were excluded from ANOVA to achieve homogeneity of variance; these can be compared to the variable values using the LSEffect (5%). ^b^ No sporulation.

**Table 3 jof-09-00238-t003:** Volatile organic compounds (VOCs) produced by *Trichoderma* sp. “*atroviride* B” isolates in the inverted plate assays with *Rhizoctonia solani*. The VOCs were sampled from the headspace by solid-phase micro-extraction 48 h after *R. solani* was introduced and were analyzed on a gas chromatograph-mass spectrometer. Putative identities were assigned based on comparisons with the NIST and Wiley databases.

VOC No	Putative Name	Synonym (or Nearest Match for Unknown)	Similarity (%)	Retention Index
1	2-Heptanone		91	900
2	2-Heptanol		95	904
3	Pentyl acetate	Acetic acid, pentyl ester	97	917
4	3-Octanone		96	989
5	(±)-Limonene ^a^	4-Isopropenyl-1-methylcyclohexene;	91	1032
6	Unknown	(4-Hydroxyphenyl)acetonitrile	83	1093
7	2-Nonanone		96	1094
8	Undecane ^a^		92	1099
9	Unknown			1118
10	Unknown	3,7-Dimethyl-6-octenal	81	1156
11	1-Butyl-4-methoxybenzene or	4-Butylphenyl methyl ether	91	1249
12	(S)-(-)-Citronellic acid	Methyl 3,7-dimethyl-6-octenoate	92	1262
13	Unknown	3-Dodecen-1-ol, acetate, (E)-	83	1285
14	2-Undecanone ^a^		96	1295
15	Tridecane ^a^		95	1298
16	Unknown			1311
17	Unknown			1399
18	α-Bergamotene	2,6-Dimethyl-6-(4-methyl-3-penten-1-yl)bicyclo [3.1.1]hept-2-ene	94	1444
19	Geranylacetone ^a^	6,10-Dimethyl-5,9-undecadien-2-one	95	1457
20	cis-β-Farnesene ^a^	(6Z)-7,11-Dimethyl-3-methylene-1,6,10-dodecatriene	91	1460
21	6-Pentyl-α-pyrone ^a^	6-Pentyl-2H-pyran-2-one	92	1476
22	(-)-Germacrene D	8-Isopropyl-1-methyl-5-methylene-1,6-cyclodecadiene	96	1496
23	Unknown	cis-γ-Bisabolene or β-Curcumene	90	1502
24	Unknown			1512
25	β-Bisabolene ^a^	(4S)-1-Methyl-4-(6-methyl-1,5-heptadien-2-yl)cyclohexene	93	1517
26	β-Sesquiphellandrene	(6S)-3-Methylene-6-[(2S)-6-methyl-5-hepten-2-yl]cyclohexene	91	1534
27	Unknown	(E)-6-Pent-1-enylpyran-2-one	85	1540
28	trans-Nerolidol ^a^	(6E)-3,7,11-Trimethyl-1,6,10-dodecatrien-3-ol	96	1570
29	Methyl trans,cis-farnesate	6,10-Dodecadienoic acid, 3,7,11-trimethyl-, methyl ester, (E)-(S)-	92	1720
30	Unknown			1995
31	Verticiol	[1R-(1R*,3E,7E,11R*,12R*)]-4,8,12,15,15-Pentamethyl- bicyclo[9.3.1]pentadeca-3,7-dien-12-ol	90	ND ^b^
32	Unknown			ND ^b^

^a^ Compound identity was corroborated by GC-MS analysis of the standard. ^b^ Not determined.

**Table 4 jof-09-00238-t004:** Mean inhibition of *Rhizoctonia solani* growth by *Trichoderma* sp. “*atroviride* B” isolates in the inverted plate assays sampled for gas chromatography-mass spectrometry analysis. The total number and quantity of VOCs detected in the headspace are listed.

Treatment	Inhibition of Pathogen Growth (%) ^a^	Total Number of VOCs	Total Quantity of VOCs (Peak Area × 10^7^)
LU521	78.2	28	7.1
LU661	67.7	28	8.8
LU132	65.8	22	5.3
LU140	65.3	22	5.3
LU633	62.5	23	5.6
LU657	59.9	24	4.1
LU584	53.7	23	3.7
LU583	46.1	19	3.3
LSD (5%)	6.7	3	1.2

^a^ Pathogen colony diameter in the treatment as a percentage of the positive control 48 h post-inoculation.

**Table 5 jof-09-00238-t005:** Volatile organic compounds (VOCs) produced by eight *Trichoderma* sp. “*atroviride* B” isolates in the inverted plate assays with *Rhizoctonia solani*.

VOC No. ^a^	Relative Peak Area (1 × 10^5^) ^b^							
	LU132	LU140	LU521	LU583	LU584	LU633	LU657	LU661
1	59.7 ± 10.7	60.5 ± 11.7	41 ± 5.9	32.9 ± 6.7	27.9 ± 4.8	98.2 ± 22.7	36.3 ± 5.8	183.0 ± 39.3
2	1.9 ± 1.2	2.6 ± 1.2	ND ^c^	ND	ND	6.3 ± 1.9	1.2 ± 0.7	10.9 ± 3.2
3	6.6 ± 2.3	4.8 ± 1.5	ND	0.2 ± 0.2	0.3 ± 0.3	0.4 ± 0.4	0.5 ± 0.3	0.7 ± 0.3
4	3.5 ± 1.9	2.5 ± 1.0	10 ± 0.5	4.0 ± 2.4	2.4 ± 0.4	6.6 ± 2.1	2.9 ± 1.4	5.7 ± 1.7
5	2.6 ± 0.2	2.8 ± 0.4	4 ± 0.3	1.2 ± 0.4	1.9 ± 0.2	2.1 ± 0.2	ND	4.0 ± 0.4
6	3.5 ± 1.0	3.2 ± 0.8	0.7 ± 0.1	1.4 ± 0.3	1.9 ± 0.6	4.2 ± 1.4	3.4 ± 0.8	7.4 ± 1.7
7	3.6 ± 0.8	3.4 ± 0.8	0.5 ± 0.1	1.8 ± 0.5	1.6 ± 0.4	4.8 ± 1.2	2.2 ± 0.3	9.9 ± 2.2
8	1.2 ± 0.5	1.2 ± 0.4	1.5 ± 0.4	0.8 ± 0.4	1.0 ± 0.3	1.1 ± 0.4	1.0 ± 0.4	1.8 ± 0.7
9	28.5 ± 4.6	22.4 ± 4.8	111.1 ± 18.2	5.4 ± 1.1	12.7 ± 2.9	17.8 ± 4.4	30.8 ± 5.8	96.5 ± 16.4
10	ND	ND	ND	ND	0.1 ± 0.1	0.1 ± 0.1	0.4 ± 0.2	0.4 ± 0.2
11	0.1 ± 0.1	ND	0.3 ± 0.09	0.1 ± 0.1	0.3 ± 0.1	ND	0.1 ± 0.1	0.5 ± 0.1
12	ND	ND	1.1 ± 0.7	ND	ND	ND	ND	ND
13	ND	ND	ND	ND	ND	ND	ND	0.4 ± 0.2
14	ND	ND	6.4 ± 0.5	0.02 ± 0.02	0.02 ± 0.02	1.4 ± 0.3	0.2 ± 0.2	2.8 ± 0.4
15	3.8 ± 1.3	4.0 ± 1.2	3.8 ± 1.7	2.2 ± 1.0	2.1 ± 0.8	2.7 ± 0.7	2.5 ± 0.6	4.8 ± 1.7
16	ND	ND	1.3 ± 0.1	ND	ND	ND	ND	1.2 ± 0.3
17	ND	0.1 ± 0.1	0.12 ± 0.1	ND	ND	ND	0.1 ± 0.1	0.2 ± 0.1
18	6.9 ± 1.3	6.5 ± 0.9	34.0 ± 9.9	1.2 ± 0.2	1.5 ± 0.3	3.0 ± 0.4	1.2 ± 0.4	24.4 ± 6.6
19	0.4 ± 0.1	0.4 ± 0.1	1.3 ± 0.1	ND	0.2 ± 0.1	0.5 ± 0.1	1.5 ± 0.2	1.9 ± 0.4
20	1.2 ± 0.3	1.2 ± 0.3	3.0 ± 0.3	ND	0.4 ± 0.3	0.6 ± 0.3	ND	3.6 ± 1.1
21	390.2 ± 59.2	394.3 ± 49.1	456.2 ± 65.5	273.5 ± 42.4	300.9 ± 51.1	398.0 ± 58.7	308.0 ± 36.6	489.6 ± 68.7
22	2.2 ± 0.6	2.1 ± 0.5	2.2 ± 0.6	1.3 ± 0.4	1.8 ± 0.5	1.1 ± 0.7	2.8 ± 1.0	3.7 ± 0.6
23	2.5 ± 0.5	3.1 ± 0.2	8.6 ± 0.8	2.0 ± 1.0	1.8 ± 0.6	3.6 ± 0.8	3.4 ± 0.9	4.2 ± 0.8
24	1.5 ± 0.3	1.6 ± 0.2	1.8 ± 0.3	0.6 ± 0.3	0.9 ± 0.3	1.1 ± 0.2	1.4 ± 0.2	2.8 ± 0.6
25	0.8 ± 0.1	0.8 ± 0.1	2.8 ± 0.3	0.3 ± 0.2	0.3 ± 0.2	0.4 ± 0.2	0.3 ± 0.2	1.4 ± 0.3
26	ND	ND	0.4 ± 0.1	ND	ND	ND	0.8 ± 0.1	ND
27	6.8 ± 0.8	7.1 ± 0.6	7.5 ± 0.6	4.2 ± 0.5	5.1 ± 0.9	7.5 ± 0.8	6.4 ± 0.3	10.6 ± 0.7
28	0.7 ± 0.1	0.7 ± 0.1	2.1 ± 0.1	0.2 ± 0.1	0.6 ± 0.2	0.5 ± 0.1	0.9 ± 0.2	1.9 ± 0.2
29	ND	ND	0.7 ± 0.0	ND	ND	ND	ND	ND
30	ND	ND	1.1 ± 0.2	ND	ND	ND	ND	0.4 ± 0.1
31	0.8 ± 0.3	0.7 ± 0.1	13.0 ± 2.7	ND	0.5 ± 0.2	0.8 ± 0.3	1.2 ± 0.1	4.6 ± 1.1
32	ND	ND	0.3 ± 0.1	ND	ND	ND	ND	ND

^a^ The names of VOC correspond to the number indicated in Table 2. ^b^ The measurements are the average and standard error from five independent samples. ^c^ Not detected.

## Data Availability

Not applicable.

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
