# Peer review of "Fungistatic Activity Mediated by Volatile Organic Compounds Is Isolate-Dependent in Trichoderma sp. “atroviride B”"

_jof, 2023, doi:10.3390/jof9020238_

Round 1

Reviewer 1 Report

Dear Authors,

I have to say that this is one of the best papers I reviewed recently. I only have a few minor comments.

Minor issues.

1., I’d note that the experimental setup in 2.5. does not control for the possible synergy between the volatile hexane and the tested antimicrobial VOCs – though you mention that alkanes have low activity, but the hexane dose added is quite huge. Would a low-volatility solvent (DMSO?) be (more) suitable?

2., Pure compounds did not differ in exp. 1 and 2 (Fig. 3-4), while an already growing culture of Trichoderma was a prerequisite of growth inhibition (L266), despite VOC emission is thought to be the causative agent being growth inhibition potential of T. isolates. What might have been in the background of this paradox?

3., Liquid injection has shown that VOCs are in the linear range of determination. Were experiments run (either now or in a referenced former work) to show that you’re not saturating the fiber at typical tested concentrations?

Best regards.

Author Response

I have to say that this is one of the best papers I reviewed recently. I only have a few minor comments.

Thanks so much.

Minor issues.

1., I’d note that the experimental setup in 2.5. does not control for the possible synergy between the volatile hexane and the tested antimicrobial VOCs – though you mention that alkanes have low activity, but the hexane dose added is quite huge. Would a low-volatility solvent (DMSO?) be (more) suitable?

Answer: Hexane was included as a control and was shown to have no biological effect on it’s own. Hence, the likelihood of it having a noticeable synergistic effect is low.

2., Pure compounds did not differ in exp. 1 and 2 (Fig. 3-4), while an already growing culture of Trichoderma was a prerequisite of growth inhibition (L266), despite VOC emission is thought to be the causative agent being growth inhibition potential of T. isolates. What might have been in the background of this paradox?

Answer: Volatiles are considered secondary metabolites; hence these are unlikely to be produced in significant quantities until after Trichoderma has started growing. In addition, some of the tested plant pathogens grow more rapidly than Trichoderma.

3., Liquid injection has shown that VOCs are in the linear range of determination. Were experiments run (either now or in a referenced former work) to show that you’re not saturating the fiber at typical tested concentrations?

Answer: We did not run any experiments testing fibre saturation. The fungal VOC concentrations measured by GC-MS were comparatively low; we assume that SPME fibre saturation would not be a problem. Furthermore, comparisons were made between fungal isolates of the same Trichoderma species; hence differences were primarily quantitative shifts within a reasonable range. Fibre saturation, which we think has not occurred, should have affected all isolates similarly.

Reviewer 2 Report

This is a very interesting paper which I read with great pleasure. The study highlighted the importance of individual Trichoderma isolates when it comes to the potential biological control options. This means that more sampling of Trichoderma is needed in all parts to determine the true potential of this great fungi.

Below are some minor suggestions I think the authors need to revise and correct the text before publishing.

Abstract, lines 16-18- The authors first mentioned that 59 isolates were tested against R.solani and therefore the “results” part of the abstract should first focus on R.solani and then on all the other fungi tested in this study. The way the authors presented this is somewhat confusing (The authors first presented “all the other fungi” (16-18), then R.solani (18-19), then again “all the other fungi” (19-20), then again R.solani (21-24), then again ”all the other fungi”-(24)???)

Also, 59 isolates were tested against R. solani, but only eight of them against other fungi. Why? Why didn’t you test all the isolates against all the fungi?

Abstract, lines 25-26. Ok, so you clearly demonstrated that there are significant differences in the VOCs profiles among the isolates of the same species. I agree. But what is the significance of this finding? Why is this so important? To whom it might be important? To agriculturalists? Foresters? Something else?

Introduction, lines 73-77 “Here we report a systematic analysis of the fungistatic activity of VOCs emitted by these four isolates along with another fifty-five T. sp. "atroviride B" isolates and the ability of VOC blends of 13 isolates with distinctive attributes to control the soil-borne fungal pathogen R. solani”. Which isolates are the authors talking about? Four isolates, then 55 isolates, then 13 isolates? 13 Why did you separate the four isolates? Why 13 isolates?

Materials- It would be very useful for the reader if the authors would transfer Supplementary Figure 1 in the main text!

Discussion- line 459- And perhaps of a pathogen strain

461-468- the importance of inhibiting sclerotia formation lies in the fact that sclerotia are resting structures that fungi use to survive unfavorable environmental conditions. By inhibiting the formation of scerotia, Trichoderma slowly but effectively “kills the enemy.”

Author Response

This is a very interesting paper which I read with great pleasure. The study highlighted the importance of individual Trichoderma isolates when it comes to the potential biological control options. This means that more sampling of Trichoderma is needed in all parts to determine the true potential of this great fungi.

Answer: Thanks so much

Below are some minor suggestions I think the authors need to revise and correct the text before publishing.

Abstract, lines 16-18- The authors first mentioned that 59 isolates were tested against R.solani and therefore the “results” part of the abstract should first focus on R.solani and then on all the other fungi tested in this study. The way the authors presented this is somewhat confusing (The authors first presented “all the other fungi” (16-18), then R.solani (18-19), then again “all the other fungi” (19-20), then again R.solani (21-24), then again ”all the other fungi”-(24)???)

Answer: The abstract has been amended to avoid this confusion.

Also, 59 isolates were tested against R. solani, but only eight of them against other fungi. Why? Why didn’t you test all the isolates against all the fungi?

Answer: To test all 59 isolates against all the pathogens is a large undertaking and we did not have the funds to do this.

Abstract, lines 25-26. Ok, so you clearly demonstrated that there are significant differences in the VOCs profiles among the isolates of the same species. I agree. But what is the significance of this finding? Why is this so important? To whom it might be important? To agriculturalists? Foresters? Something else?

Answer: The last sentence in the abstract has been amended to address this comment.

Introduction, lines 73-77 “Here we report a systematic analysis of the fungistatic activity of VOCs emitted by these four isolates along with another fifty-five T. sp. "atroviride B" isolates and the ability of VOC blends of 13 isolates with distinctive attributes to control the soil-borne fungal pathogen R. solani”. Which isolates are the authors talking about? Four isolates, then 55 isolates, then 13 isolates? 13 Why did you separate the four isolates? Why 13 isolates?

Answer: The sentence is modified to avoid any confusion.

Materials- It would be very useful for the reader if the authors would transfer Supplementary Figure 1 in the main text!

Answer: The figure is now part of the main text.

Discussion- line 459- And perhaps of a pathogen strain

Answer: The sentence has been modified accordingly.

461-468- the importance of inhibiting sclerotia formation lies in the fact that sclerotia are resting structures that fungi use to survive unfavorable environmental conditions. By inhibiting the formation of scerotia, Trichoderma slowly but effectively “kills the enemy.”

Answer: This comment has been incorporated into the discussion.